# Impact of the COVID-19 Pandemic on Tumor Stage and Pathohistological Parameters of Vulvar Cancer

**DOI:** 10.3390/jcm13144058

**Published:** 2024-07-11

**Authors:** Gilbert Georg Klamminger, Annick Bitterlich, Meletios P. Nigdelis, Laura Schnöder, Bashar Haj Hamoud, Erich-Franz Solomayer, Mathias Wagner

**Affiliations:** 1Department of General and Special Pathology, Saarland University (USAAR), 66424 Homburg, Germany; 2Department of General and Special Pathology, Saarland University Medical Center (UKS), 66424 Homburg, Germany; 3Department of Gynecology and Obstetrics, Saarland University Medical Center (UKS), 66424 Homburg, Germany; 4Saarland University Medical Center for Tumor Diseases (UTS), Saarland University (USAAR), 66424 Homburg, Germany

**Keywords:** vulvar cancer, COVID-19, histomorphological risk factors

## Abstract

**Background/Objectives**: Vulvar cancer (VC) comprises a small fraction of female neoplasms with notable high-incidence clusters among German regions. Despite a proposed impact of nationwide lockdowns in response to the COVID-19 pandemic on oncological diseases, the effect on VC staging and tumor characteristics remains yet to be resolved; therefore, analyzing pathological data from patients with squamous cell VC pre-, during, and post-COVID in a high-incidence region may offer insights into potential epidemiological and clinical trends. **Methods**: We identified a total of 90 patients who were diagnosed at the Institute of Pathology, University Hospital Saarland, between 2018 and 2023, and defined three distinct cohorts: a pre-COVID cohort (2018–2019), a COVID cohort (2020–2021), and a post-COVID cohort (2022–2023). Histomorphological data were collected from the individual patient reports and statistically analyzed using Fisher’s exact test or the Kruskal–Wallis test. **Results**: Although we found no statistically significant differences in age, T-stage, perineural infiltration, blood vessel infiltration, resection status, grading, or resection margin between our three cohorts, surprisingly, we determined a greater extent of lymphovascular infiltration (Fisher’s exact test; *p* = 0.041), as well as deeper tumor infiltration depth (Kruskal–Wallis test; *p* < 0.001) before the COVID-19 pandemic. Furthermore, we did not identify any soft indications of abnormalities in patient care within our center (unchanged status of the resection margins across all three cohorts). **Conclusions**: Our results clearly do not support a negative affection of clinical or pathobiological characteristics of VC during or after the pandemic. However, final assessments regarding the pandemic’s effect on VC require additional study approaches in various regions, preferably with future extended timeframes of a longer follow-up.

## 1. Introduction

Although accounting for just <1% of female neoplasms [1], the global incidence of vulvar cancer (VC) is varying not only regionally but also temporally, with distinct regional clusters of continued high incidence rates in Germany and South Africa, as well as a globally increasing incidence trend within the last decade(s) due to human papillomavirus (HPV) infections [2,3]. The small German state of *Saarland* stands out, with both the highest incidence nationwide as well as the most pronounced rise of the age-standardized incidence rate in general (reaching a maximum of 5.7 cases per 100,000 inhabitants), comparing the time periods 1988–2002 and 2008–2012 [3,4,5,6]. This epidemiological trend contrasts with the still disillusioning survival data of 50–70% (five-year survival) [1,7], which have not improved sufficiently in recent decades and further underline the clinical need for a broader scientific approach to VC, focusing not only on treatment options but also on pathohistological risk factors, early diagnostics, and advanced strategies of prevention [8,9,10,11].

Between 2020 and 2021, nationwide lockdowns and severe restrictions in Germany resulted as a response to the declared pandemic situation of SARS-CoV-2—severe acute respiratory syndrome coronavirus type 2—by the World Health Organization (WHO) at the beginning of March 2020. Due to such distinct restraining orders (contact bans), several cancer screening programs were temporarily canceled in a broad range of countries in order to reduce further transmission of the respiratory disease within society as much as possible and to lessen the burden on clinics, which were, at that point, heavily occupied with the emergency care of critically ill COVID-19 patients [12]. Although a final analysis of the resulting impact on oncological morbidity and mortality is still pending, it was postulated early on that strict preventive COVID measures resulted not only in a decrease in cervical cancer screenings but also overall cancer diagnoses [13,14]. A systematic review and meta-analysis by Teglia et al. analyzed the number of performed cancer screening tests (cervical cancer, breast cancer, and colorectal cancer) before and during the pandemic and determined a striking decrease, especially in cervical cancer (−51.8%) and breast cancer screening (−46.7%) [12]. An additional analysis emphasizing breast cancer screening was performed by Ng and Hamilton et al., who showed that not only breast cancer diagnosis and screening rates decreased within the pandemic but also that national, country-specific lockdown measures negatively influenced the rates of mammograms and cancer diagnosis even more [15]. At that time, in response to the proposed negative impact of acute SARS-CoV-2 infection on the timing of treatment and on the clinical course and mortality of VC patients [16], the multidisciplinary team of Garganese et al. proposed a distinct concept of action in order to enable the most adequate and personalized diagnosis and treatment under the given circumstances, considering, among others, the patients’ individual vulnerability, the COVID-19 status, the clinical cancer stage, and the house interns’ multidisciplinary tumor board decisions [17]. This seemed particularly important, given that there is currently no consensus-based prevention program to ensure that women with vulvar cancer can be treated at an early stage. In clinical practice, older women often feel ashamed to seek help, and younger women may not have vulvar cancer (VC) immediately recognized, leading to initial misdiagnosis and treatment as an infection or dermatosis.

Finally, the end of the global health emergency in 2023 will now enable a scientifically sound analysis (ex-post) of public healthcare issues and the clinical/scientific implications of the pandemic. Such retrospective evaluations should always be assessed in the specific context of the country/region of interest—a fact tragically highlighted by the reality that the COVID-19 pandemic has not only highlighted but also accelerated existing problems in several diverse national healthcare systems to a different extent [18,19]. We retrospectively analyzed pathological data (before, during, and after main COVID restrictions) of patients with squamous cell VC in a region with one of the highest incidence rates worldwide. Our aim was to assess the association between the pandemic and the risk of advanced stages of VC, depicting a change in biological tumor characteristics (tumor staging and pathological risk factors).

## 2. Materials and Methods

### 2.1. Patient Data and Data Collection

Registered patients who were diagnosed at the Institute of Pathology (University Hospital Saarland) within the years 2018–2023 were identified using our internal clinic information program and the search terms “vulvar carcinoma” and “vulvectomy”. Additionally, querying of the corresponding ICD-O codes (8085/3, 8086/3, and 8070/3) within the Saarland University Medical Center for Tumor Diseases (UTS) registry was performed. In this study, solely patients with squamous cell carcinoma of the vulva and accompanying surgery—allowing for sufficient TNM staging—were included. Exclusion criteria were defined as “high-grade dysplasia”, “bioptical diagnosis only”, “tissue-related/artifact-related insufficient pathological diagnostics”, “tumor entities of the vulva other than squamous cell carcinoma”, “recurrent tumor according to the originally assigned TNM classification within the pathological report”, and “palliative–reductive surgery”; see also Appendix A. The data collection was embedded in an ongoing research project focusing on risk factors and histological biomarkers in VC, approved by the Ethics Committee of Saarland (study identification number 249/23, approved on 7 March 2024). All data were handled in alignment with the Declaration of Helsinki [20].

After the inclusion of *n* = 90 (100%) patients, these were split into three distinct cohorts based on the year of diagnosis: a pre-COVID cohort (2018–2019), a COVID cohort (2020–2021), and a post-COVID cohort (2022–2023). The timeframe assignment of each cohort was set in accordance with the German national lockdown restrictions and the definition of the pandemic period defined by Resende et al. [21]. In the next step, individual patient reports were screened and TNM staging data were collected, including tumor stage (T1-3, tumor size, invasion, and infiltration into regional tissue structures such as the vagina and urethra), groin lymph node involvement (N1-3, the involvement of regional lymph nodes), perineural infiltration (Pn, invasion of vital tumor cells within the perineurium of peripheral nerval structures), lymphovascular space invasion (L, infiltration of neoplastic cells within lymph vessels), blood vessel infiltration (V, infiltration of neoplastic cells within blood vessels), and resection status (R, presence/absence of vital tumor cells within the surgical margin), minimum resection margin distances (proximity of definite surgical margin to closest tumor area), and grading (rating of tumor cell morphology in relation to the histomorphology of the physiologic tissue of origin) [22]. Additionally, patient age and depth of infiltration (defined by convention as the distance from the highest adjacent dermal papilla to the point of deepest infiltration) were noted. See Figure 1 for an overview of our study protocol.

### 2.2. Statistical Analysis

All data were stored in an Excel file and imported for visualization and calculation in Jamovi (Version 2.3.21.0) and GraphPad (Version 10.2.3, Boston, MA, USA). After initial data visualization, data were checked for normal distribution (Shapiro–Wilk test; *p* > 0.05 would allow for the assumption of the null hypothesis and therefore a normal data distribution). Subsequently, the Kruskal–Wallis test (including Dunn’s multiple comparisons test for a mean comparison of each group) and Fischer’s exact test were employed as non-parametric tests for hypothesis testing; α < 0.05 was considered statistically significant. Consecutively, the effect size of all available tests with subgroup analysis and α < 0.05 was estimated by calculating the value of Cohen’s d using means and standard deviations. Additionally, *p*-values were analyzed using the method of Benjamini and Hochberg to control for results incorrectly assigned as “significant” (so-called control of the false discovery rate) [23].

## 3. Results

### 3.1. Clinical Data Overview

From 90 (100%) patients with histomorphologically diagnosed squamous cell carcinoma of the vulva in 2018–2023 at the Institute of Pathology (University Hospital Saarland), 30 (33.3%) patients received their diagnosis and treatment between 2018 and 2019, 23 (25.6%) patients between 2020 and 2021, and 37 (41.1%) patients between 2022 and 2023, reflecting a decrease of overall VC diagnoses during the lockdown phase and an increase afterward within the years 2022–2023. See Table 1 for a detailed overview of the data distribution of our parameters of interest according to our a priori-defined cohorts; Appendix A provide additional information about histomorphological tumor aspects of all three cohorts.

### 3.2. Pre-, Intra-, and Post-COVID Analysis

We did not determine any statistically significant discrepancies in age (Kruskal–Wallis test; *p* = 0.483), T-stage (Kruskal–Wallis test; *p* = 0.184), or perineural infiltration (Fisher’s exact test; *p* = 0.825). Furthermore, we did not detect differences within blood vessel infiltration (Fisher’s exact test; *p* = 1.0) or tumor grading (Kruskal–Wallis test; *p* = 0.109) pre-, intra-, and post-COVID-19. A comparison of the surgical parameter resection status (Fisher’s exact test; *p* = 0.946) and resection margin distance (Kruskal–Wallis test; *p* = 0.878) did not show any differences within our three cohorts. See Table 2 for a detailed display of non-significant individual statistical assessments and their corresponding interpretations.

Interestingly, we did find divergences of our cohorts with respect to N-stage (Kruskal–Wallis test; *p* = 0.012), lymphovascular infiltration (Fisher’s exact test; *p* = 0.041), and infiltration depth (Kruskal–Wallis test; *p* < 0.001); see also Figure 2. By comparing the individual group means of the parameters “N-stage” and “infiltration depth”, we were able to confirm that the pre-COVID cohort differs significantly from both alternate two cohorts (Dunn’s multiple comparisons test: N-stage: *p* = 0.022 for pre-COVID vs. COVID and *p* = 0.047 for pre-COVID vs. post-COVID; infiltration depth: *p* = 0.007 for pre-COVID vs. COVID and *p* = 0.001 for pre-COVID vs. post-COVID), while no significant differences could be observed between the COVID and the post-COVID group (Dunn’s multiple comparisons test: N-stage: *p* > 0.999 for COVID vs. post-COVID; infiltration depth: *p* > 0.999 for COVID vs. post-COVID), see also Table 3.

A consecutive and more detailed statistical analysis proves that the aforementioned results have at least a medium-to-large effect size (infiltration depth: Cohen’s d = 1.16 for pre-COVID vs. COVID and Cohen’s d = 0.85 for pre-COVID vs. post-COVID; N-stage: Cohen’s d = 0.88 for pre-COVID vs. COVID and Cohen’s d = 0.54 for pre-COVID vs. post-COVID). In addition, the conducted analysis following Benjamini and Hochberg confirmed the suggested differences between the cohorts of our study regarding groin lymph node involvement (N-stage) and infiltrating depth, even when setting significant thresholds as *p*-values < 0.013.

## 4. Discussion

Evaluating the impact of the COVID-19 pandemic is of relevance not only in terms of health policy but also in clinical decision-making. The knowledge and interpretation of potentially deviating disease progressions post-pandemic would allow for the development and implementation of concrete measures, such as the adaptation of regular checkups due to more evolved malignancies if necessary.

It is not in doubt that necessary adjustments to the healthcare system resulting from the COVID-19 pandemic had an impact on the care of gyneco-oncological patients in 2020 and 2021, be it in terms of type and timing of oncological surgeries performed or in terms of radiation therapy—topics that have been consecutively addressed by various studies taking into account regional differences [24,25,26,27,28,29,30,31] and which resulted in specific recommendations as well as alternative management statements from various expert panels and professional societies [32,33,34,35]. Not surprisingly, a great amount of literature has already studied the impact of such statements and COVID-19-related variances in oncological diseases, albeit with sometimes contradictory findings up until now. Therefore, a focus was set on potential dynamics in tumor stages and tumor parameters during the pandemic. Despite a postulated marked decrease in cancer diagnoses with relation to sociodemographic aspects in the timeframe of the pandemic shown by the team of Han et al. in a cross-sectional study [36], significant increases in higher T-stages of head and neck cancers as well as a higher rate of progressed colorectal cancer during the pandemic were noted [37,38]. The team of Resende et al. observed an increase in advanced-stage breast cancer due to the pandemic, whereas the team of Feron Agbo et al. did not find an increase in tumor stage but rather an incline in invasive neoplasms and precancerous lesions of breast cancer in France [21,39]. Aiming to assess changes also within the post-COVID era, Hanuschak et al. determined a rather broad range of variations in clinical and pathological cancer stage dynamics of breast cancer, melanoma, and colorectal cancer pre- and post-COVID [40].

Even for vulvar cancer patients, the current literature remains inconclusive; while the team of Reid et al. reported an increase in incidence rates of vulvar neoplasms after the declared COVID-19 lockdown in a single center analysis, the group of Oymans et al. analyzed data from the Netherlands Cancer Registry and did not find a significant difference in incidence rates of VC before or during the pandemic [41,42].

Our results, which not only consider the timeframes before and during the pandemic but also take into account the post-COVID era, demonstrate that tumor stages in VC did not change due to the COVID pandemic, despite the constraint of a limited sample size. Furthermore, the analysis of the “resection margin positive/negative” and “resection margin distance” clinical parameters and surgical outcomes do not support any divergence in patient care or a loss in therapeutic-surgical treatment due to the pandemic, despite COVID-19 causing delays in elective surgeries and personnel limitations—proposing an exceptional adaptability of medical professionals even in the face of challenging circumstances [27,30]. In line with data presented by Kumar et al., we did not report any differences in morbidity and mortality rates for VC surgeries before or during COVID-19 [43]. Even though we did determine differences in lymph node involvement, lymphovascular invasion, and infiltration depth within the pre- and post-COVID cohorts, a greater extent of groin metastasis and deeper tumor infiltration was unexpectedly seen prior to the pandemic restrictions and not after the pandemic. One possible explanation for this phenomenon could be modified patient behavior due to the pandemic measures, e.g., contact restriction and lockdowns, which may have hindered low-threshold hospital visits and led to a quasi-selected patient population. In that sense, in a recent systematic review, Carbone et al. addressed rates of hospitalization and reasons for care seeking in gynecological patients and pregnant women during lockdown periods and selective control periods. Although they determined an overall increase in hospitalization rates during the pandemic (from 22.7 to 30.6%), fewer patients presented with vaginal bleeding due to gynecological reasons (7.4% vs. 9.2%) [44]. In accordance, Turner et al. demonstrated how healthcare access was impaired for women during the pandemic, highlighting a disproportionate impact for women with a history of cancer and sexual minority women [45]. To keep access to healthcare services as simple as possible, telehealth tools—which were partly implemented successfully during the pandemic—may provide future low-barrier contact with medical professionals in addition to the still-favored in-person visits [46,47].

Yet another possible rationale of our data presented—contrary to the aforementioned reasoning—could have been the psychological impact of the pandemic on patients causing exactly the opposite effect. Specifically, patients’ generally increased vigilance regarding health awareness may explain the superior compliance and earlier realization of medical (screening) examinations, leading to earlier disease detection at a less advanced stage of the disease during and after the pandemic. In favor of the latter argument, higher levels of anxiety and distress were reported during the pandemic for patients affected by gynecological cancers [48,49]. Although such findings raise the necessary attention to increase psychological support and strengthen individual psychoeducational tools, their final effects and implications in cancer screening remain to be resolved; this can be remedied in future studies by analyzing a larger observation period.

Last but not least, the ongoing scientific discourse on the pathophysiological interplay between COVID-19 as a viral disease and oncological effects also has to be mentioned, in order to consider oncological diseases such as VC and their pathohistological diagnosis in their full clinical and scientific context. Patients with a systematic oncological treatment (such as chemotherapy or radiotherapy) are not only at a higher risk of more serious COVID-19 infections due to hampered cellular host immune responses due to a gain of pro-inflammatory IL-17-producing T cells and cytotoxic T cells as well as a decline of CD4 T cells and CD8 T cells [50,51], but also a link on a molecular level between cancer and COVID-19 has been proposed. In a review, Ghosh et al. highlight commonly shared features of both pathological conditions on an epigenetic, genetic, and proteomic level. Examples of this are protease CTSL/B, which is not only involved in biochemical processes of viral cell entry but is also amplified in different neoplasms, and ACE2 enzyme, which seems to be overexpressed in some carcinomas [52]. Within the scope and according to the design of this study, pathophysiological and microbiological aspects of the relationship between COVID-19 and VC were not considered in detail; however, their potential context should be properly considered in order to put all epidemiological and clinical examinations into a meaningful perspective.

### Limitations

In contrast to the existing literature on COVID-19 and VC [41,42,43], our study analyzes the impact of the pandemic on pathomorphological risk factors. Limitations of the study include its unicentric nature, which cannot take into account regional differences, as well as its lack of a deeper pathophysiological analysis. However, the aforementioned topic clearly remains beyond the scope of this study. Final assessments about the pandemic’s impact on VC need to be drawn in accordance with additional multicenter studies, which aim to elucidate the impact of the pandemic in a global approach, also considering the impact of sociodemographic factors and potential variabilities in healthcare access.

## 5. Conclusions

The presented study compares the clinical and pathobiological characteristics of 90 patients with histomorphologically diagnosed VC pre- (2018–2019), during (2020–2021), and post-COVID (2022–2023). Our statistical results illustrate that these characteristics of VC were not impaired either during the pandemic or post-COVID. Following our analysis, a consecutive negative impact of the pandemic on prognosis may not be deduced—if anything, our data suggest that the quality of surgical cancer care remained consistent despite the unprecedented burden on the healthcare system caused by COVID-19, highlighting the ability of medical professionals to uphold the highest guideline-adhering standards of patient care despite challenging circumstances. Within clinical routines, a close monitoring of patients who have been diagnosed or treated during the pandemic according to current guidelines should maximize patients’ outcomes.

## Figures and Tables

**Figure 1 jcm-13-04058-f001:**
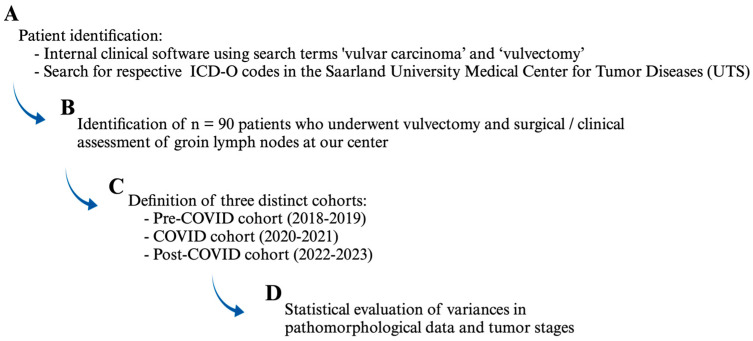
After initial patient identification and selection (**A**,**B**) in accordance with our inclusion and exclusion criteria (see text), three distinct cohorts were defined (**C**), pathomorphological variables were individually collected, and statistical analysis was performed (**D**). Figure 1 was created with Biorender.com.

**Figure 2 jcm-13-04058-f002:**
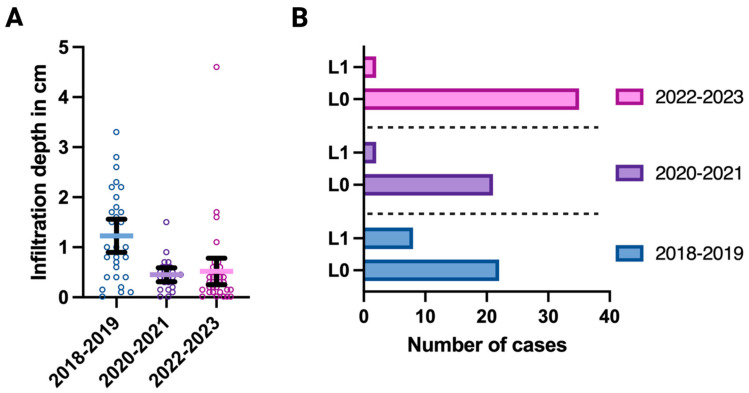
(**A**) Optical display of all tumors’ individual (colored small circles) depth of infiltration (in cm; y-axis) according to each a priori-defined cohort (*x*-axis), showing significant group differences (Kruskal–Wallis test; *p* < 0.001). (**B**) Visualization of lymphovascular space invasion (L1 = vital tumor cells within lymphovascular spaces, L0 = no distinct lymphovascular infiltration; *y*-axis) according to each a priori-defined cohort (colored bars), showing significant group differences (Fisher’s exact test; *p* = 0.041). The total numbers of cases are presented on the *x*-axis. Figure 2 was created with Biorender.com.

**Table 1 jcm-13-04058-t001:** Baseline characteristics of selected histopathological tumor parameters within our defined patient cohorts. * Single patients were excluded due to tissue-related/artifact-related insufficient pathological diagnostics.

Parameter of Interest	Pre-COVID Cohort *n* = 30	COVID Cohort *n* = 23	Post-COVID Cohort *n* = 37
Median age (IQR) year	71.5 (63.5–81)	63 (54–79)	71 (60–81)
Pn positive	4 (13.3%)	2 (8.7%)	3 (8.1%)
V positive	1 (3.3%)	0 (0%)	1 (2.7%)
L positive	8 (26.7%)	2 (8.7%)	2 (5.4%)
T-stage: T1a	3 (10%)	4 (17.4%)	9 (24.3%)
T-stage: T1b	23 (76.7%)	18 (78.3%)	26 (70.3%)
T-stage: T2	4 (13.3%)	1 (4.3%)	2 (5.4%)
N-stage: N0	19 (63.3%)	21 (91.3%)	32 (86.1%)
N-stage: Micrometastasis	1 (3.3%)	1 (4.3%)	0 (0%)
N-stage: N1a	1 (3.3%)	1 (4.3%)	2 (5.6%)
N-stage: N1b	1 (3.3%)	0 (0%)	0 (0%)
N-stage: N1c	0 (0%)	0 (0%)	0 (0%)
N-stage: N2a	0 (0%)	0 (0%)	0 (0%)
N-stage: N2b	0 (0%)	0 (0%)	0 (0%)
N-stage: N2c	8 (26.7%)	0 (0%)	3 (8.3%)
Grading: G1	1 (3.3%)	3 (13%)	6 (16.2%)
Grading: G2	12 (40%)	11 (47.8%)	18 (48.6%)
Grading: G3	17 (56.7%)	9 (39.1%)	13 (35.1%)
Resection status: R0	24 (80%)	19 (82.6%)	29 (78.4%)
Resection status: R1	6 (20%)	4 (17.4%)	8 (21.6%)
resection margin distance * mean (SD) cm	0.32 (0.2)	0.27 (0.2)	0.29 (0.2)
infiltration depth mean (SD) cm	1.23 (0.9)	0.45 (0.3)	0.52 (0.8)

IQR: interquartile range, L: lymphovascular infiltration, N-stage: involvement of groin lymph nodes, Pn: perineural infiltration, SD: standardized deviation, T-stage: tumor stage, V: vascular infiltration.

**Table 2 jcm-13-04058-t002:** Comparison of non-significant histopathological tumor parameters according to the cohorts defined. Pn: perineural infiltration, R: resection margin status, T-stage: tumor stage, V: vascular infiltration.

Parameter of Interest	Statistical Analysis	Result	Interpretation
Age	Kruskal–Wallis test	*p* = 0.483 (χ^2^ value = 1.46)	medians do not vary significantly
T-stage	Kruskal–Wallis test	*p* = 0.184 (χ^2^ value = 3.39)	medians do not vary significantly
Pn	Fisher’s exact test	*p* = 0.825	no significant associations within the contingency table
V	Fisher’s exact test	*p* = 1.0	no significant associations within the contingency table
Grading	Kruskal–Wallis test	*p* = 0.109 (χ^2^ value = 4.44)	medians do not vary significantly
R	Fisher’s exact test	*p* = 0.946	no significant associations within the contingency table
Resection margin distance	Kruskal–Wallis test	*p* = 0.878 (χ^2^ value = 0.26)	medians do not vary significantly

Pn: perineural infiltration, R: resection margin status, T-stage: tumor stage, V: vascular infiltration.

**Table 3 jcm-13-04058-t003:** Comparison of selected histopathological tumor parameters according to the cohorts defined.

Parameter of Interest	Statistical Analysis	Result	Interpretation
N-stage	Kruskal–Wallis test	*p* = 0.012 (χ^2^ value = 8.82)	medians do vary significantly
N-stage—“2020–2021 vs. 2018–2019”	Dunn’s multiple comparisons test	*p* = 0.022 (z value: 2.7)	significant associations
N-stage—“2022–2023 vs. 2018–2019”	Dunn’s multiple comparisons test	*p* = 0.047 (z value: 2.4)	significant associations
N-stage—“2022–2023 vs. 2020–2021”	Dunn’s multiple comparisons test	*p* > 0.999 (z value: 0.6)	no significant associations
L	Fisher’s exact test	*p* = 0.041	significant associations within the contingency table
Infiltration depth	Kruskal–Wallis test	*p* < 0.001 (χ^2^ value = 17.97)	medians do vary significantly
Infiltration depth—“2020–2021 vs. 2018–2019”	Dunn’s multiple comparisons test	*p* = 0.007 (z value: 3.0)	significant associations
Infiltration depth—“2022–2023 vs. 2018–2019”	Dunn’s multiple comparisons test	*p* = 0.001 (z value: 4.1)	significant associations
Infiltration depth—“2022–2023 vs. 2020–2021”	Dunn’s multiple comparisons test	*p* > 0.999 (z value: 0.6)	no significant associations

L: lymphovascular infiltration, N-stage: involvement of groin lymph nodes.

## Data Availability

Please contact the corresponding author for individual solutions.

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
