# Peer review of "Impact of the COVID-19 Pandemic on Tumor Stage and Pathohistological Parameters of Vulvar Cancer"

_jcm, 2024, doi:10.3390/jcm13144058_

Round 1
Reviewer 1 Report
Comments and Suggestions for Authors
I read with attention the present article.
- Even though the SARS-CoV-2 pandemics had an impact on diagnosis and treatment of gynecological diseases in the years 2020-2022, access to hospital facilities has nowadays become increasingly effortless;
- Authors have not clearly presented the association between the pandemics and histopathology of vulvar cancer;
- Introduction and Discussion sections should be completely revised, explaining whether there is a relationship between SARS-CoV-2 infection and vulvar cancer in terms of microbiological pathways.
Author Response
Comment 1:
I read with attention the present article.
Response 1:
We thank the reviewer for the positive assessment of our work and the helpful comment that allowed us to improve our manuscript.
Comment 2:
Even though the SARS-CoV-2 pandemics had an impact on diagnosis and treatment of gynecological diseases in the years 2020-2022, access to hospital facilities has nowadays become increasingly effortless;Authors have not clearly presented the association between the pandemics and histopathology of vulvar cancer; Introduction and Discussion sections should be completely revised, explaining whether there is a relationship between SARS-CoV-2 infection and vulvar cancer in terms of microbiological pathways.
Response 2:
We thank the review for the constructive feedback. We extensively worked on our discussion part and added the necessary information about the path of physiological connection between COVID-19 and cancer. Additionally, we also added more references and information about the association of COVID-19 and vulvar cancer.
Furthermore, in accordance with distinct request from reviewer 2 and reviewer 3 we re-arranged the structure of our material and methods as well as results part and performed additional analysis.
Reviewer 2 Report
Comments and Suggestions for Authors
The article tries to identify if there are any notable changes in which concerns vulvar cancer incidence and staging in three distinct time-frames, each of them of two years length: a pre-Covid two years period, a Covid one between 2020-2021, and a post Covid period. As factor determining the extension of the disease they analyzed TNM staging with tumor size, invasion into surrounding structures, lymph node positivity, perineural infiltration, lymphovascular space invasion, blood vessel in filtration, as well as resection status, minimum resection margin distances and tumor grading. The authors found no statistical significant differences between these three analyzed periods of time, underlining again the tremendous efforts of the health-care providers, despite the overwhelming pandemic, to succeed in offering the same access and medical care to their patients. But unexpectedly they found statistical significant differences in which concern limp-node involvement, lymphovascular invasion, and infiltration depth between the pre- and post-Covid timeframes.
Despite the low number of cases (90) in a high incidence region, which weakens its statistical power and impacts the results, the authors manage to offer a six year time-frame picture of vulvar cancer staging.
I would be interested in using a method to quantify the magnitude and the clinical impact of your findings, like confidence intervals or effect sizes and use a correction method to control type I errors, adjustments for multiple hypothesis testing.
Row 54 – “the” beginning of March
Row 64 – decrease especially in cervical cancer
Please rephrase the last phrase of the introduction section just before Material and Methods section – is too long, make more phrases instead of such a long one to enhance clarity.
What does “Single patients were excluded ???” form Table 1
Comments on the Quality of English Languagerevision needed
Author Response
Comment 1: The article tries to identify if there are any notable changes in which concerns vulvar cancer incidence and staging in three distinct time-frames, each of them of two years length: a pre-Covid two years period, a Covid one between 2020-2021, and a post Covid period. As factor determining the extension of the disease they analyzed TNM staging with tumor size, invasion into surrounding structures, lymph node positivity, perineural infiltration, lymphovascular space invasion, blood vessel in filtration, as well as resection status, minimum resection margin distances and tumor grading. The authors found no statistical significant differences between these three analyzed periods of time, underlining again the tremendous efforts of the health-care providers, despite the overwhelming pandemic, to succeed in offering the same access and medical care to their patients. But unexpectedly they found statistical significant differences in which concern limp-node involvement, lymphovascular invasion, and infiltration depth between the pre- and post-Covid timeframes. Despite the low number of cases (90) in a high incidence region, which weakens its statistical power and impacts the results, the authors manage to offer a six year time-frame picture of vulvar cancer staging.
Response 1:
We thank the reviewer for this overall positive assessment of our work and hope to address the points raised to match the expectations for a consideration of publication in JCM. In accordance also with the request from reviewer 1 and reviewer 3, we restructured our results part, and added more information to our discussion part.
Comment 2: I would be interested in using a method to quantify the magnitude and the clinical impact of your findings, like confidence intervals or effect sizes and use a correction method to control type I errors, adjustments for multiple hypothesis testing.
Response 2:
In order to evaluate effect sizes and control type one errors, we additionally added a more sophisticated analysis (Cohen's d and analysis of p-values using the method of Benjamini and Hochberg); please refer also to our revised material and methods part, as well as our revised results part.
Comment 3: Row 54 – “the” beginning of March
Response 3:
Thanks, corrected.
Comment 4: Row 64 – decrease especially in cervical cancer
Response 4:
Thanks, corrected.
Comment 5: Please rephrase the last phrase of the introduction section just before Material and Methods section – is too long, make more phrases instead of such a long one to enhance clarity.
Response 5:
Thanks, corrected.
Comment 6: What does “Single patients were excluded ???” form Table 1.
Response 6:
We thank the review for this comment. As we corrected in the revised version of our manuscript, the “exclusion of single patients” refers to the analysis of the tumor depth infiltration. As depicted in Supp. Table 3, one vulvectomy specimen did not allow for analysis of depth of infiltration and was therefore excluded.
Reviewer 3 Report
Comments and Suggestions for Authors
Congratulations to the authors for choosing this topic.
Although the topic is very interesting, the article is not well structured.
Introduction:
- Line 40 the incidence of vulvar cancer is increasing worldwide and not just in South Africa
- One of the main reasons for the rising incidence of vulvar cancer in younger women is HPV…you should mention that
- Lines 62 – 69 - prevention programs are not directly related to vulvar cancer and the aim of your study. Instead, you should write more about vulvar cancer – there is no prevention program and the big problem with women with vulvar cancer is that they do not see a doctor right away…the older women are sometimes ashamed to seek help and in younger women vulvar cancer is sometimes not recognized and treated as an infection or dermatosis.
- At the end you should clearly state the aim of the study.
Materials and methods
- You should clearly state the inclusion and exclusion criteria of your study
- Figure 1: The font of the figure is not the same as that of the text
Results
- line 142 – Is it a planocelular subtype of vulvar carcinoma?
- tables 1 and 2 are not well organized
- there is no type of surgery, no tumor diameter, no information on SNB or lymphadenectomy,…
Discussion
- The discussion is not well organized
- The discussion is general and does not focus on vulvar cancer
- You should clearly point out the limitations of the study, of which there are many
Author Response
Comment 1:
Congratulations to the authors for choosing this topic. Although the topic is very interesting, the article is not well structured.
Response 1:
We thank the reviewer for the positive assessment of our idea and the thoughtful and helpful comments that helped us to improve the structure of our manuscript. In order to provide the best possible insight for the interested reader, we have also explicitly rearranged/revised the Material and Methods and Results sections and extensively worked on our Discussion part.
Introduction:
Comment 2:
- Line 40 the incidence of vulvar cancer is increasing worldwide and not just in South Africa
- One of the main reasons for the rising incidence of vulvar cancer in younger women is HPV…you should mention that
Response 2:
Thank you, this is correct and may be of interest for the reader. As requested, we added/corrected the missing information, see line 38, 39.
Comment 3:
- Lines 62 – 69 - prevention programs are not directly related to vulvar cancer and the aim of your study. Instead, you should write more about vulvar cancer – there is no prevention program and the big problem with women with vulvar cancer is that they do not see a doctor right away…the older women are sometimes ashamed to seek help and in younger women vulvar cancer is sometimes not recognized and treated as an infection or dermatosis.
Response 3:
We added the missing information, see line 83-88.
Comment 4:
- At the end you should clearly state the aim of the study.
Response 4:
We thank the reviewer for the hint and accordingly changed the phrasing, see line 98, 99.
Materials and methods
Comment 5:
- You should clearly state the inclusion and exclusion criteria of your study
Response 5:
We do agree with the reviewer that clear assessable inclusion and exclusion criteria of a study are crucial. We therefore added a file with Supplementary Material and Supp. Table 1 where all a priori defined inclusion and exclusion criteria are listed in addition to their mention within the main text.
Comment 6:
- Figure 1: The font of the figure is not the same as that of the text
Response 6:
Thanks! We corrected the font of figure 1!
Results
Comment 7:
- line 142 – Is it a planocelular subtype of vulvar carcinoma?
Response 7:
We would like to use this point-by-point answer to address the interesting terminology raised by the reviewer: As the term "planocellular carcinoma" is not commonly used nationally and is furthermore not used in the current WHO classification of female genital tumors ("blue books") in the vulva chapter, we have decided to dispense with the term "planocellular carcinoma" in favor of a terminology that is presumably more accessible to the general public and " simply" use the term "squamous cell carcinoma" instead. At this it is important for us to use consistent terminology throughout the manuscript.
Comment 8:
- tables 1 and 2 are not well organized
Response 8:
We thank the reviewer for the feedback and agree that the abundance of content displayed within both tables could be optimized in the way of visualization. For this reason, we have divided the information content of the tables into a total of three tables, see also the new tables 1 - 3.
Comment 9:
- there is no type of surgery, no tumor diameter, no information on SNB or lymphadenectomy,…
Response 9:
We do agree with the reviewer, that additional data may serve useful fot the interested reader. Taking into account the predominant histopathological focus of our manuscript we added Supp. Table 2/3/4 within our new Supplementary Material containing additional information about the tumor biology.
Discussion
Comment 10:
- The discussion is not well organized
- The discussion is general and does not focus on vulvar cancer
Response 10:
We thank the reviewer for the hint. In accordance especially with the urgent request of reviewer 1 we added more information about the connection of COVID-19 and cancer on a pathophysiological level and therefore improved the storyline of our discussion.
Additionally, we added new references and information especially about the VC and COVID-19, see line 415-419 and 434-436.
Comment 11:
- You should clearly point out the limitations of the study, of which there are many
Response 11:
As requested by the reviewer, we added a paragraph “4.1. Limitations” to our Discussion, see line 500-509.
Round 2
Reviewer 1 Report
Comments and Suggestions for Authors
The Authors have responded to all the comments made to the previous version of the manuscript.
Author Response
We thank the reviewer for the positive feedback.
Reviewer 2 Report
Comments and Suggestions for Authors
I am pleased with the correction.
Author Response

(The authors gave the same response as above.)

Reviewer 3 Report
Comments and Suggestions for Authors
Congratulations to the authors. The article has been significantly improved.
Just one small recommendation: You should add the abbreviations at the end of the tables.
Author Response
We thank the reviewer for the comments and the positive feedback.
As requested, we added the abbreviations to the end (down under) the tables.